# Repeated Cross-Scale Structure-Induced Feature Fusion Network for 2D Hand Pose Estimation

**DOI:** 10.3390/e25050724

**Published:** 2023-04-27

**Authors:** Xin Guan, Huan Shen, Charles Okanda Nyatega, Qiang Li

**Affiliations:** 1School of Microelectronics, Tianjin University, Tianjin 300072, China; 2School of Electrical and Information Engineering, Tianjin University, Tianjin 300072, China

**Keywords:** hand pose estimation, RGB image, self-occluded, multi-layer features, feature fusion

## Abstract

Recently, the use of convolutional neural networks for hand pose estimation from RGB images has dramatically improved. However, self-occluded keypoint inference in hand pose estimation is still a challenging task. We argue that these occluded keypoints cannot be readily recognized directly from traditional appearance features, and sufficient contextual information among the keypoints is especially needed to induce feature learning. Therefore, we propose a new repeated cross-scale structure-induced feature fusion network to learn about the representations of keypoints with rich information, ’informed’ by the relationships between different abstraction levels of features. Our network consists of two modules: GlobalNet and RegionalNet. GlobalNet roughly locates hand joints based on a new feature pyramid structure by combining higher semantic information and more global spatial scale information. RegionalNet further refines keypoint representation learning via a four-stage cross-scale feature fusion network, which learns shallow appearance features induced by more implicit hand structure information, so that when identifying occluded keypoints, the network can use augmented features to better locate the positions. The experimental results show that our method outperforms the state-of-the-art methods for 2D hand pose estimation on two public datasets, STB and RHD.

## 1. Introduction

Hand pose estimation is one of the most important topics in natural human–computer interaction. Moreover, it has become a key technology in human–computer interaction (HCI), virtual reality (VR), sign language recognition, medical treatments, and interactive entertainment [1,2,3,4,5,6], which can help humans to communicate with machines in a more natural way. However, due to the complex articulation and high flexibility of hand fingers, hand pose estimation remains challenging [7]. To address these difficulties, it is important to adopt multi-view images of the hand [8,9]. Unfortunately, a multi-view acquisition system requires expensive hardware and strict environmental configurations. In recent years, alternative schemas with RGB-D images with depth information have been proposed for hand pose estimation due to the popularity of low-cost depth sensors [10,11,12]. However, this method has limited scenarios and is easily affected by the surrounding environment, so it cannot be widely used. Compared with the data format above, RGB images can be readily available via low-cost optical sensors, such as webcams and smartphones, which makes hand pose estimation based on RGB images widely popular.

Even for 3D hand pose estimation directly from RGB images [13,14,15,16,17,18], most proposed methods still need to estimate 2D hand poses first, and then upgrade the estimated 2D poses to 3D [15,16,17]. With regard to 2D hand images, it is crucial to locate the occluded joints that are pervasive in various hand poses. To tackle this challenge, the methods are mainly divided into two categories: one is to mine the hand image features and locate the keypoints by learning the keypoint information hidden in the hand features, and the other is to mine the geometric relationship between the keypoints of the hand and locate the keypoints by learning the physical constraint relationship between them. YG Wang et al. [7] constructed a cascade network, including mask prediction and attitude estimation, to provide some soft constraints to hand-feature learning. S Kyeongeun et al. [19] proposed a new implicit semantic data augmentation method that can generate additional training samples in the feature space and improve the performance of hand pose estimation. TH Pen et al. [20] proposed an optimized convolutional pose machine (OCPM) for accurately estimating hand poses. OCPM uses the CBAM attention module and information processing module to extract hand features more effectively, leading to higher performance. Moreover, YG Wang et al. [21] iteratively used the interested region of the hand as feedback information to strengthen hand feature representation by an encoder–decoder framework. DY Kong et al. [22] proposed an adaptive graph model network (AGMN) that combines deep learning with a graph model to better learn the geometric relationship between hand closing points, thus improving the accuracy of hand pose estimation. Additionally, YF Chen et al. [23] proposed a new nonparametric structure regularization machine (NSRM) for hand attitude estimation. NSRM learns the representation of hand composition and structure to regularize the confidence graph of keypoints in a nonparametric way, effectively strengthening the structural consistency of the predicted hand posture. Although the overall hand constraint is enhanced by extracted feature methods, such as mask, attitude, and ROI, the occluded keypoints are in need of information on the adjacent keypoints on the inside of the hand to assist the characterization. the graph model and structural constraint methods are usually carried out after feature representation, but the feature learning of occluded joints has not been effectively guided, and there is still room for improvement in its effectiveness.

When examining variable keypoints, traditional feature learning networks work well for clear visible keypoints. For obscure ones, additional perceptual fields can capture the features, and for invisible points, more contextual information is necessary. Therefore, we propose a novel repeated cross-scale fusion network to learn semantic-induced features in hand images. Our network is divided into two parts: GlobalNet and RegionalNet. In GlobalNet, four different scales of features are generated through ResNet50, providing rich neighborhood information among keypoints. Then, we merge the deep semantic features with the shallow features, one by one, to roughly locate the keypoints of the hand. In RegionalNet, GlobalNet generates four features of different scales in parallel. These features undergo multiple cross-scale convolutional fusions, allowing the deep features to continuously exchange information with the shallow features, allowing the network to learn rich deep semantic and contextual information, which is beneficial for locating occlusion keypoints.

We show the efficiency of our proposed method on two public datasets: STB [24] and RHD [15]. Our approach significantly outperforms the current state-of-the-art algorithms on both datasets. Qualitative results show that our method can locate hand keypoints more efficiently when self-occlusion is present.

The main contributions of this work are:We propose a new and effective hand pose estimation network, which integrates the global feature pyramid network (GlobalNet) and then further refines the feature fusion network (RegionalNet). It can provide rich information through the relationship between features at different levels of abstraction.We explore the impacts of various feature fusion methods that contribute to the localization of occlusion keypoints, and further demonstrate that the effective use of deep-level information is highly beneficial for the mining of occlusion keypoints.Our model achieves high accuracy and can mine the keypoints of hand occlusion more effectively than other methods.

## 2. Related Work

The progress on 2D hand pose estimation from a single RGB image and the multi-scale feature fusion related to our method is as follows.

### 2.1. Hand Pose Estimation

Hand pose estimation can be divided into two categories: regression-based and detection-based methods. The former directly regresses the hand image to the coordinates of keypoints position and were mainly used in the early stage [25,26,27]. This approach makes the training and forward computing fast but ignores the spatial information on the feature map, resulting in poor spatial generalization. On the other hand, the latter is inspired by human pose estimation [28] and are widely used in hand pose estimation [29,30,31]. This method predicts a heatmap for each keypoint as the intermediate representation of the keypoint. The maximum value on the heatmap corresponds to the coordinates of the corresponding keypoint, where the ground truth of the heatmap is a two-dimensional Gaussian distribution centered on keypoints [32]. The heatmap adds a gradual distribution process to the target location, which helps the network find the gradient descent more smoothly and alleviates overfitting in case of incorrect labeling.

Compared to the human body, the hand always covers a small area in the entire RGB image, making hand pose estimation more challenging due to the high degree of freedom of hand joints, similar skin colors, and self-occlusion. In addition, due to the lack of depth information and susceptibility to factors such as illumination, existing hand pose estimation methods based on RGB images still need help in locating the occlusion keypoints efficiently and with satisfactory precision. Regarding robustness to environmental changes, YK Li et al. [29] proposed a self-disentanglement method, which decomposes the monochrome hand image into the representative features of the hand pose and the complementary appearance features of the image. However, it is still challenging to locate the occluded keypoints precisely through only the adjacent appearance features. To learn the correlation between keypoints, N Santavas et al. [30] proposed an end-to-end lightweight network that can understand the global constraints and correlations among keypoints through attention-enhanced reverse bottleneck blocks. This improves the accuracy of hand pose estimation by sharing “collective knowledge” among subsequent layers. DY Kong et al. [31] proposed a new structure called the rotation invariant mixed graph model network (RMGMN), which combines the graph model and depth convolutional neural network in a novel way, and can explicitly model the spatial constraints between hand joints, thus improving the prediction performance of occluded keypoints. However, the pairwise parameters in the GM are fixed, and cannot adapt to the characteristics of each input image. Additionally, adopting belief propagation reasoning increases the complexity of the model.

### 2.2. Method Using Multi-Layer Features

Multi-scale feature fusion is widely used in the field of human/hand pose estimation [33,34,35,36,37,38,39], because different parts of the human/hand require feature maps of different scales to represent the keypoint locations. Suppose the keypoints are detected only through the feature map of the last layer. Therefore, the multi-scale network structure considers the information exchange and feature extraction between feature maps of different scales at the design’s beginning and then fuse the features of different scales to obtain the final output. For example, the feature pyramid network (FPN) [33] predicts images of different sizes based on predictions. The cascaded pyramid network (CPN) [34] is a modified version of FPN that converts low-resolution features into high-resolution features through an upsampling process. Hourglass [35] continuously downsamples and upsamples to capture feature information at different resolutions. The bisected hourglass network [36] improves the original hourglass network by enabling it to output both keypoint heatmaps and hand masks. Simple baseline [37] combines upsampling and convolution parameters in a simpler way with the deconvolution layer based on ResNet, without using jump layer connections. On the other hand, HRNet [38] effectively exchanges information between multiple-resolution feature channels while maintaining a high-resolution feature representation of the image to obtain a highly accurate high-resolution output. Additionally, HandyPose [39] gradually learns rich spatial and contextual information in the feature space through an advanced multi-level waterfall module, which improves the accuracy of hand pose estimation.

Unlike the human body, which has obvious distinctions between different limbs, the hand has the same skin color in different parts, making hand pose estimation much more challenging than human body pose estimation; moreover, occluded keypoints are difficult to mine using the appearance features of the hand. Thus, we received some inspiration from the above methods to make the network focus more on deep-level feature learning and used deep features to assist the shallow features for keypoint mining.

## 3. Method

Given a colored image of the hand $I\in R^{w*h*3}$, our task is to detect the hand pose by using the convolutional neural network (CNN). We express the 2D hand pose (Pk) as the position of the joint point in the image, i.e., $p_{k}=\left(x_{k},y_{k}\right)$ (*k* is the number of labeled keypoints in the hand; here, we set k=21), where $p_{k}$ represents the 2D pixel coordinates of the *k*-th keypoint in image I.

An overview of the proposed network structure, which includes two subnetworks, GlobalNet and RegionalNet, is illustrated in Figure 1. The purpose of GlobalNet is to roughly locate the joint points and build the loss function in the first stage as intermediate supervision. The subsequent RegionalNet is responsible for predicting the joint points more precisely.

### 3.1. GlobalNet

GlobalNet adopts a pyramid structure with four scale features, C2, C3, C4, and C5, from ResNet50. Shallow features, such as C2 and C3, have a higher spatial resolution but lower-level semantics, while deep features, such as C4 and C5, have the opposite characteristics. A jump connection is added to the FPN, as shown by the dashed line in Figure 1, between the input and output nodes at the same scale to fuse more features without adding too much computational cost. The heatmaps output by GlobalNet are used to learn the approximate location of keypoints and serve as intermediate supervision. In CPN, the output of GlobalNet will perform upsampling and 3 × 3 convolution operations on four different scale features to generate four keypoint heatmaps. As far as we know, there is no significant difference between these four heatmaps. Therefore, we will upsample and merge the four scale features and then output one keypoint heatmap through 3 × 3 convolution operations, which will not only fulfill the intermediate supervision function of GlobalNet but also reduce the number of network parameters.

### 3.2. RegionalNet

After learning from GlobalNet, the network roughly located the keypoints. However, the prediction accuracy of the network did not achieve satisfactory results because the feature information learned by GlobalNet was minimal. Therefore, we attached a RegionalNet behind GlobalNet to obtain a more detailed prediction.

RegionalNet connects the four-scale features in parallel, as shown in Figure 1. At the same time, the features from deep to shallow are gradually increased to form a new level. Therefore, the features of the parallel subnet at a certain level are composed of the features of the same scale from the previous level and other parallel features of different scales. The specific fusion mode is shown in Figure 2. The four-way feature graphs in GlobalNet are added with the features of the corresponding scale at the beginning of each stage in RegionalNet. For features with the same resolution, the bottleneck blocks are stacked directly to maintain self-resolution. For the feature that needs to be increased in the resolution, an upsampling operation is used to add and fuse the feature with the corresponding resolution. For the feature that needs to reduce the resolution, we use 3 × 3 convolution and add/fuse with the corresponding resolution feature. Finally, the features of all scales are concatenated to compute the final heatmap prediction.

RegionalNet uses cross-scale convolutional fusion to fuse information of different scales. It is not a simple upsampling step that aggregates low-level and high-level information together. In this process, each level feature receives information repeatedly from other parallel features, allowing for the integration of rich spatial and semantic information. This greatly increases the ability to mine occluded keypoints.

### 3.3. Loss Functions

**Estimated loss in GlobalNet.** For GlobalNet, we apply L2 loss to 2D heatmaps of *k* keypoints, and calculate the estimated loss of GlobalNet. The loss function $L_{G}$ is defined as follows:(1)$$L_{G}=\sum_{j=1}^{k}{∥h_{j}^{G\;}-h_{j}^{gt}∥}_{2}$$
where $h_{j}^{G\;}$ and $h_{j}^{\mathrm{gt}\;}$ represent the estimated and ground truth heatmaps, respectively, for the *j*-th keypoint in GlobalNet. The ground truth heatmaps for the *j*-th keypoint are generated by applying a 2D Gaussian centered at its ground truth 2D keypoint location.

**Estimated loss in RegionalNet.** For RegionalNet, we use the method of online hard keypoint mining. We sort the loss values of 2D heatmaps of the *k* keypoints and select only the top *m* keypoints with larger losses for backward gradient propagation. The loss function is defined as follows:(2)$$L_{R}=\sum_{j=1}^{m}{∥h_{j}^{R\;}-h_{j}^{gt}∥}_{2}$$
where $h_{j}^{R\;}$ and $h_{j}^{\mathrm{gt}\;}$ represent the RegionalNet estimated and ground truth heatmaps of the *j*-th keypoint, respectively. *m* represents the number of selected heatmaps with the largest loss values (m<=k).

**Total Loss of Network.** The total loss $L_{\mathrm{total}\;}$ of our entire network is defined as follows:(3)$$L_{\mathrm{total}\;}=L_{G\;}+L_{R}$$

## 4. Experiments

In this section, several experiments are conducted to verify the performance of the proposed method; ablation experiments were carried out in STB [24] and RHD [15]. The experimental results show that our method can increase estimation precision compared with the state-of-the-art methods.

### 4.1. Experimental Settings

**Dataset and Evaluation Metrics**. Our proposed method is a 2D hand pose estimation technique based on a single RGB image. Therefore, traditional hand pose datasets based on depth images, such as MSRA [32] and NYU [40], are not suitable for our approach. Therefore, we chose two open datasets, STB and RHD, both of which include RGB images of human hands and the position coordinates of 2D keypoints. Among them, RHD is a synthetic dataset, which consists of 39 different actions of 20 different hands, including 41,258 training samples and 2728 test samples, and the image pixels in the dataset are 320 × 320. We split RHD into 38,530 training images, 2728 validation samples, and 2728 test samples. The STB dataset is a real image dataset collected from different cameras; it contains 18,000 images and can be divided into 6 scenes. Each scene contains two RGB images and a depth image with the same action at different positions. The image pixels in the dataset are 640 × 480. We split STB into 12,000 training images, 3000 validation samples, and 3000 test samples. Following [13], we transferred the root joint in STB from the palm to the wrist to make it consistent with RHD. In these two datasets, we mirrored the left hands to the right hands.

We used the probabilistic-predicted keypoint (PCK) [9] to quantitatively compare the predicted results. PCK calculates the proportion of the normalized distance between the predicted keypoints and their corresponding ground truth that is less than the set threshold σ. For the *k*-th hand keypoint $p_{k}$, we used $PCK_{\sigma }^{k}$ to represent it, and approximate it to Formula (Equation 4) on the verification dataset D.
(4)$${\mathrm{PCK}}_{\sigma }^{k}=\frac{1}{∥D∥}\sum_{D}\left({∥p_{k}^{pt}-p_{k}^{gt}∥}_{2}\leq \sigma \right)$$
where $p^{pt}$ represents the position of predicted keypoints, $p^{gt}$ represents the position of real keypoints. We used a normalized threshold σ, ranging from 0 to 1, with respect to the size of the hand’s bounding box.

**Implementation Details.** Our model was implemented using PyTorch. Before the images were inserted into the model, all images were resized to 256 × 256, and a heatmap with a size of 64 × 64 was generated for each joint. The Adam optimizer was used for training, in which the batch training size was set to 32, and the initial learning rate was set to 5 × 10${}^{-4}$. In the 100th iteration, the learning rate was reduced to half of the original one, and the training stopped at the 150th iteration. Moreover, dropout and early stop strategies were used during the training process to avoid overfitting problems in the network. Our network parameters were randomly initialized, and no other external datasets were trained in advance. All our experiments were run on a desktop with NVIDIA 2080Ti GPU.

For a comprehensive assessment, we used CPN and HRNet as the baselines of our method. For CPN, we chose CPN50 and CPN101 structures. For HRNet, we tested the results of both HRNet-W32 and HRNet-W48 structures. In addition, we trained and tested CPN, HRNet, and our method under the same experimental settings, and all of the models were trained from scratch.

### 4.2. Ablation Experiment

The proposed subsection aims to validate the effectiveness of our network through various experiments. Unless specified otherwise, all experiments in this subsection are conducted on the STB and RHD datasets. The input size for all models is 256 × 256.

**Comparison with baselines.** In Table 1 and Table 2, we compare the performance of our method with two baseline methods, CPN and HRNet. In order to prove that our method can effectively solve the problem of identifying occluded keypoints, we list the recognition accuracy of the 21 keypoints in the tables. At the same time, Figure 3 shows the corresponding schematic diagram of hand joint points.

As shown in Table 1 for the STB dataset, the wrist and thumb locations have the lowest recognition accuracy compared to the other 19 joints in the 21 joints identified by the six methods. The reason may be that the wrist and thumb positions are similar to the nearby skin texture without more apparent details. In our method, GlobalNet can roughly identify the approximate location of each node, while the recognition accuracy of keypoints is greatly improved by adding RegionalNet to learn high-level semantic information. Compared with the baseline method, our method can achieve the best recognition of each keypoint. In addition, the accuracy of recognition is further improved by introducing the OHKM method.

As RHD is a synthetic dataset, its appearance and posture distribution are different from those of real datasets. For example, the hand images in RHD do not have a knuckle pattern, and the difference between each joint is not obvious. Therefore, it is only possible to distinguish different joint points by simple features, such as the adjacent appearance of hands, which is why the accuracy of our GlobalNet test in Table 2 is much higher. At the same time, CPN and HRNet achieve higher accuracy than our method, because CPN captures more deep semantic information through its RefineNet, and CPN101 can obtain more information due to its deeper layers compared to CPN50. However, our RegionalNet uses a multi-scale fusion method through repeated cross-line convolution. Each feature from deep to shallow continuously receives information from other parallel scales, allowing the network to learn richer semantic and spatial information of keypoints to improve the accuracy for occluded keypoints. Furthermore, the accuracy is further improved when OHKM is applied in RegionalNet.

The PCK performance of our proposed model on two standard datasets (STB and RHD) is presented in Table 3. As observed, our model consistently outperforms the two baseline methods on both datasets. Moreover, we included some representative images to demonstrate the prediction results, as shown in Figure 4 and Figure 5. Our model can effectively detect occluded keypoints, such as the little finger part of the third row of images and the middle finger part of the sixth row of images. Our model performed significantly better than the baseline methods when tested on the RHD dataset.

As can be seen in Figure 4 and Figure 5, our model can recognize relatively complex hand poses, but there are still some instances of recognition errors in extreme cases, as shown in Figure 5. Similar to the last picture in Figure 6, the fingers in the photographed images are blurred due to the rapid movement of the hand. While our method may not accurately identify the keypoints of the middle finger, its performance is still better than the baseline method. Additionally, we found that when the fingers are close together, our method cannot accurately identify the keypoints of each finger, such as the little finger and the ring finger in the penultimate picture, but its performance is still better than that of the baselines.

**Design Choices of RegionalNet.** Here, we compare different design strategies of RegionalNet, as shown in Table 4. The output of our pyramid based on GlobalNet is compared using the following implementations:Prediction using only the heat map output from GlobalNet.A structure similar to HRNet is connected behind GlobalNet, starting with a high-resolution sub-network as the first stage and gradually adding high-resolution to low-resolution sub-networks, one by one, to form more stages, connecting multi-resolution sub-networks in parallel.In contrast to the structure in (2), the low-resolution subnetwork is added to the high-resolution subnetwork, one by one, starting from the low-resolution subnetwork as the first stage, to form more stages, and connecting the multi-resolution subnetworks in parallel, as shown in Figure 1.

Finally, a convolution layer is attached to generate the score maps for each keypoint.

We observed that both the models in (2) and our method achieved better accuracy compared to keypoint prediction using GlobalNet only. Our method can be compared with (2), revealing that deep semantic information of the network is more beneficial for predicting hand keypoints compared to high-resolution information.

**Online Hard Keypoint Mining.** Here, we discuss the losses used in our network. In detail, the loss function of GlobalNet is the L2 loss of all labeled keypoints, while the second stage attempts to learn hard keypoints, i.e., we only penalize the loss of the first *m* (*m*<=*m*) keypoints in *k* (the number of labeled keypoints in a hand, e.g., 21 in the STB dataset). Table 5 shows the impact of *m*. In the case of m=11, the performance of the second stage achieves the best results of the balanced training between hard and simple keypoints.

### 4.3. Comparison with the State-of-the-Art Methods

We compared our method with other state-of-the-art methods and reported the PCK test results with a threshold value of 0.02 in Table 6. It can be observed that on the STB dataset, our method outperforms SRHandNet by 0.16 in terms of PCK, outperforms [15] by 0.06, outperforms [41] by 0.23, outperforms NSRM by 0.19, outperforms [35] by 0.22, outperforms InterHand by 0.06, and outperforms [30] by 0.09. The method also outperforms SRHandNet, [41], NSRM and [35] on the RHD dataset by 0.22.

In Table 7, we present a comparison of the execution times, the number of model parameters, and the number of GFlops for all methods on a laptop configured with an NVIDIA 1650 GPU. The reported execution times are the forward inference times of the model without any acceleration. It is evident that our method can execute at 34.86 frames per second, indicating superior performance compared to the other methods. Although our method has a larger number of model parameters relative to the other methods, the GFlops of the network is not high, which further demonstrates the superiority of our proposed network.

## 5. Conclusions

In this paper, we propose a new 2D hand pose estimation network, the repeated cross-scale structure-induced feature fusion network, which is designed to identify occluded keypoints. Our network consists of two modules, GlobalNet and RegionalNet, where RegionalNet connects features of different scales in parallel and uses cross-layer convolution to fuse features of different scales many times. Throughout the entire process, deep features are always involved in the exchange of information between other scales, allowing the network to learn deeper semantic information and improve the accuracy and mining ability of occluded keypoints. The experimental results on the STB and RHD public datasets demonstrate that the proposed repeated cross-scale fusion network outperforms state-of-the-art methods.

## Figures and Tables

**Figure 1 entropy-25-00724-f001:**
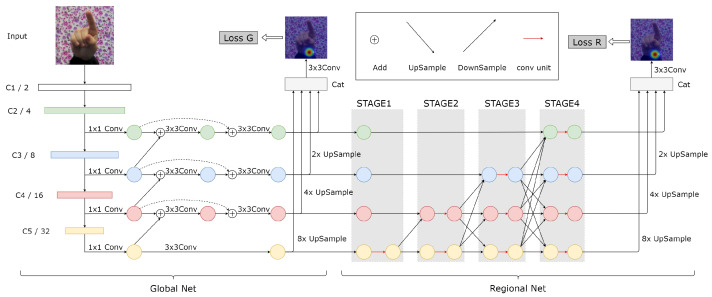
Overview of our architecture. It comprises two modules: GlobalNet and RegionalNet.

**Figure 2 entropy-25-00724-f002:**
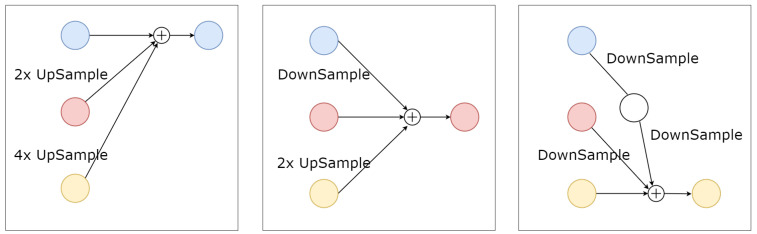
Schematic diagram of the fusion method. This explains how the exchange unit fuses features of different scales, from left to right. The upsample adopts the upsampling method of 1 × 1 convolution, and the downsample adopts the downsampling method of 3 × 3 convolution with a stride of 2.

**Figure 3 entropy-25-00724-f003:**
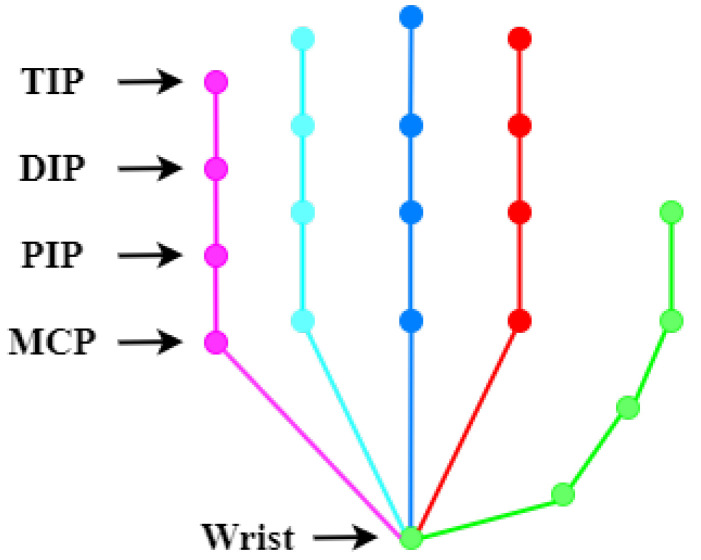
Schematic diagram of hand joint points.

**Figure 4 entropy-25-00724-f004:**
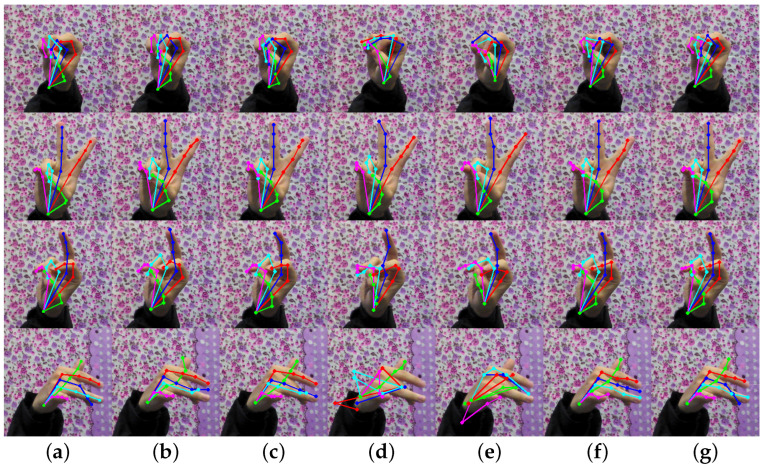
Comparison results on STB. Where (**a**) is the true label, (**b**) is the result of the CPN50 estimation, (**c**) is the result of the CPN101 estimation, (**d**) is the result of the HRNet-W32 estimation, (**e**) is the result of the HRNet-W48 estimation, (**f**) is the result of our method’s estimation without OHKM, and (**g**) is the result of our method’s estimation with OHKM.

**Figure 5 entropy-25-00724-f005:**
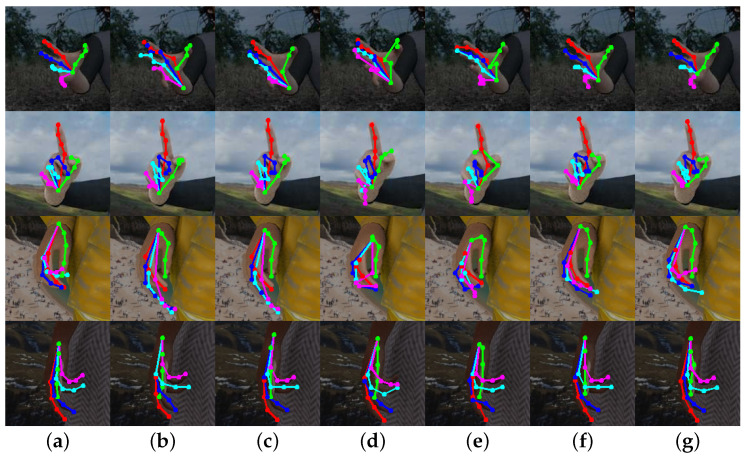
Comparison results on RHD. Where (**a**) is the true label, (**b**) is the result of the CPN50 estimation, (**c**) is the result of the CPN101 estimation, (**d**) is the result of the HRNet-W32 estimation, (**e**) is the result of the HRNet-W48 estimation, (**f**) is the result of our method’s estimation without OHKM, and (**g**) is the result of our method’s estimation with OHKM.

**Figure 6 entropy-25-00724-f006:**
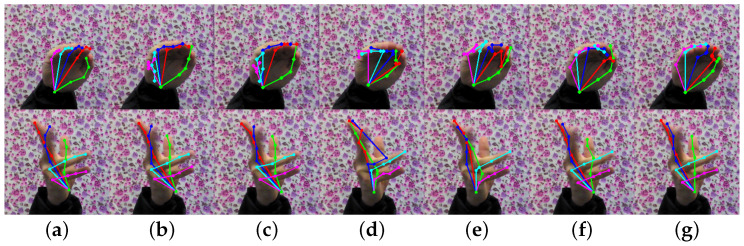
Some cases of poor performance. Where (**a**) is the true label, (**b**) is the result of the CPN50 estimation, (**c**) is the result of the CPN101 estimation, (**d**) is the result of the HRNet-W32 estimation, (**e**) is the result of the HRNet-W48 estimation, (**f**) is the result of our method’s estimation without OHKM, (**g**) is the result of our method’s estimation with OHKM.

**Table 1 entropy-25-00724-t001:** Detailed results of testing on the STB dataset. Ours * indicat s the results without online hard keypoints mining.

		CPN50	CPN101	HRNet-W32	HRNet-W48	Ours *	Ours
Wrist	Wrist	0.939	0.940	0.913	0.915	0.932	0.944
thumb	MCP	0.969	0.969	0.945	0.958	0.969	0.973
	PIP	0.961	0.963	0.932	0.935	0.967	0.972
	DIP	0.964	0.964	0.947	0.947	0.970	0.976
	TIP	0.982	0.982	0.947	0.965	0.983	0.989
index	MCP	0.989	0.991	0.964	0.978	0.991	0.995
	PIP	0.978	0.983	0.972	0.976	0.979	0.986
	DIP	0.990	0.990	0.970	0.984	0.990	0.991
	TIP	0.982	0.982	0.946	0.94	0.984	0.992
middle	MCP	0.993	0.994	0.972	0.988	0.994	0.997
	PIP	0.982	0.982	0.962	0.975	0.984	0.984
	DIP	0.971	0.986	0.943	0.969	0.975	0.981
	TIP	0.973	0.981	0.949	0.950	0.983	0.993
ring	MCP	0.989	0.992	0.968	0.979	0.989	0.995
	PIP	0.984	0.984	0.970	0.978	0.983	0.987
	DIP	0.987	0.986	0.969	0.964	0.985	0.987
	TIP	0.979	0.979	0.956	0.954	0.981	0.990
little	MCP	0.984	0.986	0.965	0.965	0.984	0.995
	PIP	0.984	0.984	0.973	0.972	0.984	0.984
	DIP	0.981	0.986	0.973	0.971	0.982	0.991
	TIP	0.976	0.979	0.949	0.961	0.980	0.987

**Table 2 entropy-25-00724-t002:** Detailed results of testing on the RHD dataset. Ours * indicates the results without online hard keypoints mining.

		CPN50	CPN101	HRNet-W32	HRNet-W48	Ours *	Ours
Wrist	Wrist	0.854	0.846	0.779	0.794	0.866	0.894
thumb	MCP	0.951	0.949	0.90	0.932	0.943	0.977
	PIP	0.945	0.948	0.888	0.916	0.948	0.970
	DIP	0.933	0.933	0.885	0.913	0.935	0.958
	TIP	0.903	0.914	0.860	0.885	0.929	0.936
index	MCP	0.971	0.975	0.920	0.943	0.971	0.989
	PIP	0.952	0.949	0.898	0.929	0.956	0.970
	DIP	0.933	0.934	0.889	0.913	0.935	0.953
	TIP	0.910	0.918	0.855	0.885	0.919	0.936
middle	MCP	0.978	0.981	0.928	0.951	0.983	0.991
	PIP	0.934	0.942	0.869	0.901	0.941	0.972
	DIP	0.907	0.916	0.861	0.888	0.921	0.950
	TIP	0.855	0.854	0.813	0.830	0.878	0.905
ring	MCP	0.975	0.975	0.925	0.944	0.980	0.990
	PIP	0.955	0.958	0.910	0.924	0.964	0.979
	DIP	0.922	0.926	0.868	0.904	0.922	0.956
	TIP	0.847	0.854	0.796	0.831	0.866	0.902
little	MCP	0.958	0.964	0.892	0.923	0.966	0.979
	PIP	0.921	0.921	0.860	0.892	0.9921	0.944
	DIP	0.901	0.912	0.847	0.889	0.915	0.933
	TIP	0.857	0.854	0.790	0.828	0.863	0.884

**Table 3 entropy-25-00724-t003:** PCK performance on two common datasets. Ours * indicates the results without online hard keypoints mining.

PCK	0.01	0.02	0.03	0.04	0.05
STB					
CPN50	0.2084	0.5931	0.7886	0.9280	0.9614
CPN101	0.2475	0.6572	0.8382	0.9466	0.9773
HRNet-W32	0.1709	0.5299	0.7298	0.9013	0.9483
HRNet-W48	0.1722	0.5303	0.7292	0.9084	0.9531
Ours *	0.2493	0.6581	0.8366	0.9521	0.9789
Ours	0.2673	0.6856	0.8483	0.9558	0.9858
RHD					
CPN50	0.2410	0.5951	0.7570	0.8787	0.9211
CPN101	0.2494	0.6018	0.7612	0.8878	0.9293
HRNet-W32	0.1703	0.4830	0.6534	0.8077	0.8683
HRNet-W48	0.1904	0.5178	0.6889	0.8394	0.8966
Ours *	0.2461	0.5968	0.7653	0.8944	0.9312
Ours	0.29	0.6628	0.8180	0.9174	0.9508

**Table 4 entropy-25-00724-t004:** Comparison of models of different design choices of RegionalNet (the average PCK@0.02 score).

Models	STB	RHD
Models in (1)	0.4382	0.4280
Models in (2)	0.5206	0.5193
ours	0.6856	0.6628

**Table 5 entropy-25-00724-t005:** Comparison of the number of different hard keypoints in online hard keypoint mining (average PCK@0.02 score).

M	9	11	13	15	17	19	21
PCK	0.663	0.685	0.683	0.680	0.676	0.664	0.658

**Table 6 entropy-25-00724-t006:** Comparison of hand keypoint estimation across different datasets (average PCK@0.02 score).

Model	STB	RHD
SRHandNet [21]	0.53	0.46
Model in [15]	0.63	0.58
Model in [41]	0.46	0.44
NSRM [33]	0.50	0.49
Model in [35]	0.47	0.45
InterHand [42]	0.63	0.61
Model in [30]	0.60	0.57
Ours	0.69	0.66

**Table 7 entropy-25-00724-t007:** Comparison of reasoning times (frames-per-second—FPS) and computational complexities of different pose estimation models.

Model	FPS	Size (MB)	Calculations (GFlops)
SRHandNet [21]	22.18	71.90	−
Model in [15]	17.14	101.4	53.47
Model in [41]	16.38	49.34	55.38
NSRM [33]	15.51	139.55	102.76
Model in [35]	20.37	8.73	25.35
InterHand [42]	21.34	541.7	−
Model in [30]	3.98	7.13	2.32
Ours	34.86	120.20	12.01

## Data Availability

The data presented in this study are available in the article.

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
