# Peer review of "Repeated Cross-Scale Structure-Induced Feature Fusion Network for 2D Hand Pose Estimation"

_entropy, 2023, doi:10.3390/e25050724_

Round 1

Reviewer 1 Report

Authors have done research for 2D pose estimation.  Overall the research is interesting. I do have a complain for the research. It looks like the accuracy is relatively low in improvement from the earlier results.  hand pose estimation is an old topic. I understand it could be challenging to get very high improvement. But what the authors have done only improved the accuracy to a tiny margin.  

I am also not sure whether the improvement is without overfitting or among others since there is not really much I justify based on what authors presented such as details regard to cross-validation, feature removal, regularization, early stopping.

Reviewer 2 Report

The authors propose a 2D hand pose estimation method that try to establish its superiority over other existing ones using standard metrics on two public datasets. The authors should note that the broader research topic they are trying to advance has received a great deal of attention from researchers for decades. In that context, it seems the background and related work section of the submitted manuscript is lacking, with obvious omissions like https://doi.org/10.1109/JSEN.2020.3018172 among possible several others. Thus, this reviewer also has concerns on whether the state-of-art methods that the authors choose to compare their proposed method with were selected after an exhaustive search of (recent) published literature in prominent venues. Additionally, there does not seem to be any discussion in the submitted manuscript on the execution time of the proposed method compared to the state-of-art. This limits the impact of the submitted manuscript as hand pose estimation is a classic problem that very often require real-time performance on compute constrained hardware.

Reviewer 3 Report

The authors propose a novel method for hand tracking with RGB images based on Neural Networks. The manuscript is well presented, the method is very detailed and the experiments are extensively complete. The overall quality of the manuscript is very high, thus, almost no changes are required. The obtained scores are so interesting that could be very useful to access the code to allow other researchers to run experiments with the proposed method also on other datasets.

Round 2

Reviewer 2 Report

The authors have adequately addressed this reviewer's concerns for the most part. The only minor comment this reviewer has is adding an explanation or analysis of why the "Model in [15]" has an abnormally high performance score on the RHD dataset.
